# Fast Multi-Distance Time-Domain NIRS and DCS System for Clinical Applications

**DOI:** 10.3390/s24227375

**Published:** 2024-11-19

**Authors:** Marco Nabacino, Caterina Amendola, Davide Contini, Rebecca Re, Lorenzo Spinelli, Alessandro Torricelli

**Affiliations:** 1Dipartimento di Fisica, Politecnico di Milano, Piazza Leonardo da Vinci 32, 20133 Milan, Italy; caterina.amendola@polimi.it (C.A.); davide.contini@polimi.it (D.C.); rebecca.re@polimi.it (R.R.); 2Consiglio Nazionale delle Ricerche, Istituto di Fotonica e Nanotecnologie, Piazza Leonardo da Vinci 32, 20133 Milan, Italy; lorenzo.spinelli@ifn.cnr.it

**Keywords:** diffuse correlation spectroscopy, time-domain near-infrared spectroscopy, hybrid device, multi-channel device, fast acquisition, software correlator

## Abstract

We have designed and built an improved system for combined Time-Domain Near-Infrared Spectroscopy (TD NIRS) and Diffuse Correlation Spectroscopy (DCS) measurements. The system features two independent channels, enabling TD NIRS and DCS acquisition at short and long source-detector distances to enhance depth sensitivity in layered tissues. Moreover, the device can operate at fast acquisition rates (up to 50 Hz) to monitor hemodynamic oscillations in biological tissues. An OEM (Original Equipment Manufacturer) TD NIRS device enables stable and robust acquisition of photon distribution of time-of-flight. For the DCS signals, the use of a time tagger and a software correlator allows us flexibility in post-processing. A user-friendly GUI controls TD NIRS data acquisition and online data analysis. We present results for the system characterization on calibrated tissue phantoms according to standardized protocols for performance assessment of TD NIRS and DCS devices. In-vivo measurements during rest and during vascular occlusions are also reported to validate the system in real settings.

## 1. Introduction

In recent years, Diffuse Optics (DO) applied to the biomedical field has gained popularity. Other than its inherent safety and non-invasiveness, the advantages of using light to probe biological tissue include relatively deep penetration inside the investigated sample, high temporal resolution and the possibility of continuous bedside monitoring.

Among DO techniques, Time-Domain (TD) Near InfraRed Spectroscopy (NIRS) harnesses the temporal broadening and attenuation of short pulses of red and near-infrared light to measure tissue optical properties (absorption coefficient, *µ_a_*, and reduced scattering coefficient, *µ_s_*′) [1]. From these, absolute concentrations of oxygenated hemoglobin (HbO_2_), deoxygenated hemoglobin (HHb), total hemoglobin (tHb = HbO_2_ + HHb) and tissue oxygen saturation (StO_2_ = HbO_2_/tHb) can be calculated. Examples of applications of TD NIRS include muscle oxidative metabolism assessment [2], spectroscopic tumor characterization [3], and investigation of functional brain activation [1].

Diffuse Correlation Spectroscopy (DCS) is a newer DO technique that exploits long-coherence light sources to quantify blood flow [4]. As light diffuses inside the sample, it becomes scattered by moving red blood cells, creating a time-varying speckle pattern on the detection plane. By measuring the temporal autocorrelation of a single speckle, it is possible to retrieve the Blood Flow Index (BFI) of the investigated sample. At the clinical level, DCS applications include measuring microvascular blood flow in skeletal muscles, both in resting-state conditions to assess endothelial function [5] and during exercise [6], or to investigate tissue perfusion in peripheral vascular diseases [7]. Extensive research has been carried out on the application of DCS to the brain [8], where it has been used for continuously monitoring cerebral blood flow in comatose patients [9] and measuring cerebral autoregulation in ischemic stroke patients [10].

Whereas low acquisition rates (about 1 Hz) are sufficient for the aforementioned applications [11], carrying out measurements at higher sampling rates (20–40 Hz) allows for the measurement of the pulsatile waveform of blood flow, which provides a non-invasive means of estimating intracranial pressure (ICP) [12]. ICP Monitoring is crucial in several neuropathologies; however, currently, the only clinically accepted methods are invasive, limiting their applicability to patients in critical conditions [13]. Non-invasive ways of estimating ICP could make it possible to extend neurologic care to a broader range of patients.

In order to extract the BFI from DCS measurements, knowledge of the tissue optical properties is needed. Therefore, DCS devices should also feature some way of measuring them, either by using multiple wavelengths and source-detector separations [14] or by incorporating Time-Domain or Frequency-Domain NIRS in the same instrument [15,16,17,18]. This kind of approach also enables the measurement of additional functional parameters, such as the cerebral metabolic rate of oxygen consumption [19], thus providing an extensive all-optical characterization of the investigated tissue.

In this work, we present a hybrid TD NIRS and DCS device for clinical applications, thoroughly characterizing it through measurements on calibrated phantoms following established protocols, as well as in-vivo to replicate clinical settings. Improving upon an analogous system we previously developed [15], which could only carry out TD NIRS measurement at a single source-detector separation, the new instrument features two separate channels for both TD NIRS and DCS, allowing better depth discrimination in layered tissues [20,21]. Moreover, the maximum achievable acquisition rate has been increased from 1 Hz to 50 Hz, enabling the detection of fast hemodynamic signals. Finally, while the previous device used a digital correlator for DCS, the new instrument features a time tagger to store the raw photon data and a software algorithm to compute the autocorrelation [22]. Accessing the raw signal offers more flexibility in data analysis [23], including pre-processing and the possibility of adopting different integration times for the same measurement.

## 2. Materials and Methods

### 2.1. Device Description

The device, together with its block scheme, is shown in Figure 1a,b. The TD NIRS module is a state-of-the-art OEM (Original Equipment Manufacturer) device (PIONIRS s.r.l., Milan, Italy) [24] featuring two laser sources operating at 685 nm and 830 nm, respectively. The light pulses pass through two variable optical attenuators before being coupled into two graded-index optical fibers (100/125 µm core/cladding diameter, Fiberoptic Systems Inc., Simi Valley, CA, USA), bundled together to create a single injection spot. Diffused light is collected by two 2 mm bundles of graded-index optical fibers (100/140 µm core/cladding diameter, Fiberoptic Systems Inc., Simi Valley, CA, USA) and directed to two silicon photomultipliers (SiPM). Two Time-Correlated Single-Photon Counting (TCSPC) boards allow reconstruction of the Distribution of Time-Of-Flight (DTOF) curves.

The DCS module employs as a source a highly coherent continuous-wave (CW) laser diode, operating at 785 nm (iBeam Smart, TOPTICA Photonics AG, Munich, Germany). The emitted light is sent to a 50/50 beam splitter and coupled into two step-index optical fibers (400/425 µm core/cladding diameter, Fiberoptic Systems Inc., Simi Valley, CA, USA) to create two separate injection points. The backscattered light is collected by four step-index single-mode optical fibers (4.4/125 µm core/cladding diameter, Fiberoptic Systems Inc., Simi Valley, CA, USA) and sent to four single-photon avalanche diodes (SPCM-AQRH, Excelitas Technologies Corp., Waltham, MA, USA). A multi-channel time-tagging board (TimeTagger20, Swabian Instruments GmbH, Stuttgart, Germany) allows us to store the photon arrival times for each channel.

The whole instrument is contained inside a (47 × 41 × 17) cm^3^ case and is operated via an external PC. An in-house software (written in C# language) allows synchronized TD NIRS and DCS acquisition as well as real-time data analysis and visualization through a user-friendly interface.

The injection and collection of optical fibers are hosted in a custom-made flexible 3D-printed optical probe, whose sketch is shown in Figure 1c. The probe was designed specifically for in-vivo measurements: the fibers run parallel to the sample plane, and the flexibility guarantees solid adherence to the skin. Both in injection and detection, light is deflected by 90° using 3.5 mm prisms. As shown in Figure 1c, two separate injection spots are employed: the first one hosts a bundle containing the two TD NIRS injection fibers and one DCS injection fiber, while the second one hosts the other DCS injection fiber. Splitting the DCS injection in two spots allows us to duplicate the signal level while complying with safety limits; indeed, according to EN-60825-1, the maximum injectable power in human tissue for a 785 nm CW laser, 0.39 NA fibers and 7 mm fiber-to-tissue distance is 28 mW. The probe also features two separate detection spots, at 15 mm (short distance) and 25 mm (long distance), respectively, from both injections. The detection site at the long distance hosts a detection fiber bundle coupled to one TD NIRS detector and three DCS detectors, while, at the short distance, one TD NIRS detector and one DCS detector are used, respectively. To equalize the signal detected at the two distances across the two TD NIRS wavelengths, the probe incorporates a combination of a 0.8 OD neutral density (ND) filter (Kodak, Rochester, NY, USA) and a black-printed transparent film.

### 2.2. Characterization Measurements

In this section, the different measurements carried out to characterize the device will be presented. Three well-established protocols in DO (BIP [25], MEDPHOT [26], nEUROPt [27]) were employed for the characterization of the TD NIRS module, and further relevant measurements were performed to obtain a comprehensive characterization of the DCS module.

#### 2.2.1. BIP Protocol

The first protocol followed was the Basic Instrumental Performance (BIP) protocol [25], whose aim is to assess the performance of TD optical devices. The protocol focuses on measuring the basic characteristics of the instrument, namely, the following:The relevant parameters of the source component: illuminated area on the sample surface, maximum power delivered to the sample.The responsivity of the detection system, defined as the ratio between the detected photon count rate for a given input radiance *L* and the radiance itself:
*s*(*λ*) = *N*_tot_/[*t*_meas_*L*(*λ*)].(1)

Here, *s* is the responsivity, *λ* is the wavelength, *N*_tot_ is the total photon count at the detector and *t*_meas_ is the duration of the measurement. To measure the radiance, a calibrated phantom with known diffuse transmittance *κ* was employed so that the radiance could be calculated from the measured input power *P*_in_:*L*(*λ*) = *P*_in_(*λ*) *κ*(*λ*).(2)

The responsivity was then evaluated according to Equation (1).

To evaluate *s* with a good signal-to-noise ratio, 20 repeated measurements with 500 ms integration time per wavelength were taken, and the resulting photon counts were summed over the repetitions.
Differential nonlinearity (DNL) of the timing electronics is defined as the nonuniformity of the time channel width of the TCSPC board. To assess the DNL, a 5-min measurement with 500 ms integration time was taken in response to a continuous light signal. The histograms for each repetition were then summed to obtain the DNL with a good signal-to-noise ratio.Instrument Response Function (IRF), both as a whole (shape) and through its relevant parameters (signal-dependent background, center of mass and Full-Width-at-Half-Maximum (FWHM)). The IRF for each source-detector pair was measured by facing the injection and detection fibers, with a thin layer of diffusive material and ND filters in between. The signal-dependent background was characterized by calculating the afterpulsing ratio *R*_AP_:
*R*_AP_ = (*N*_bkg_
*− N*_dark_)*T/*(*N*_tot_ Δ*t*),(3)
where *N*_bkg_ is the background of the IRF, *N*_dark_ is the background in a “dark” measurement (without laser source), *T* is the laser period, *N*_tot_ is the total counts in the IRF (after background subtraction) and Δ*t* = 9.77 ps is the time channel width. Both the IRF and the dark measurement consisted of 20 repetitions with an integration time of 500 ms per wavelength.

The behavior of the IRF over time was assessed by taking a 12-h stability measurement (integration time: 500 ms per wavelength), from the device being powered on, in order to investigate the warm-up times of the whole instrument and detect any unwanted drifts of the IRF parameters.

#### 2.2.2. MEDPHOT Protocol

The second protocol followed was the MEDPHOT protocol [26], which enables the assessment of the capability of the TD NIRS module to retrieve optical parameters of homogeneous turbid media. It is based on measuring a kit of 32 solid resin phantoms with known optical properties (*µ_a_* ranging from ~0 to ~0.42 cm^−1^ and *µ_s_*′ from ~5 cm^−1^ to ~20 cm^−1^ at 690 nm) and comprises the following measurements:Linearity and accuracy: The retrieved optical parameters for all phantoms were compared to the ones measured on the same set of phantoms by a state-of-the-art laboratory TD-NIRS system by the same manufacturer. The goal of this measurement was to assess the ability of the TD NIRS module to follow variations without distortions (linearity) and to retrieve accurate values of the optical parameters (accuracy). Ten repeated measurements were taken on each phantom, with an integration time of 500 ms per wavelength.Noise: The B2 phantom (*µ_a_* ≅ 0.07 cm^−1^, *µ_s_*′ ≅ 10 cm^−1^) was measured, repeatedly varying the photon counts to assess their effect on the retrieved DTOFs. For each measurement, the integration time was set so as to meet the target photon count, and 10 repeated measurements were taken. Only the detection channel at 2.5 cm source-detector separation was used. The coefficient of variation (CV) of the optical parameters was then calculated as the ratio between the standard deviation and the mean value over the 10 repetitions.Stability: A 14 h measurement with an integration time of 1 s was carried out on a liquid phantom to assess the behavior of the TD NIRS module over time. The phantom was composed of distilled water, Intralipid and black India ink, and was crafted to mimic optical properties typical of biological tissue (*µ_a_* ≅ 0.1 cm^−1^, *µ_s_*′ ≅ 10 cm^−1^) [28]. The extracted quantities were the optical parameters and the photon counts of the DTOFs.Reproducibility: Over the course of several days, several measurements were carried out on the same liquid phantom to assess the ability of the device to reproduce the same results in the same experimental conditions. The phantom used was a non-degradable water-based solution of polydisperse microparticles (HemoPhotonics S. L., Barcelona, Spain). Each measurement consisted of 30 repetitions with an integration time of 1 s, and the optical parameters obtained on different days were compared.

#### 2.2.3. nEUROPt Protocol

The final protocol followed was the nEUROPt protocol [27], whose aim is to assess the capability of fNIRS instruments to detect changes in the optical properties of the brain while rejecting the contribution of extracerebral tissues. The following measurements were carried out:Depth sensitivity: A solid phantom with a movable absorbing inclusion was employed. The source and detection fibers were placed at a 2.5 cm distance so that the inclusion was at the midpoint between them. The inclusion depth (*z* coordinate) was then varied from 0 to 30 mm in steps of 2 mm, and for each depth, 10 repetitions were acquired with an integration time of 500 ms per wavelength. The resulting DTOFs were divided into time gates 500 ps long, and, for each gate *k* and depth *z*, the contrast *C_k_* was then calculated as follows:
*C_k_*(*z*) = − log[*N_k_*(*z*)/*N_k_*_,ref_],(4)
where *N_k_* is the number of photon counts in gate *k*, summed over the 10 repetitions, and *N_k_*_,ref_ is the reference number of photons detected when no inclusion is present.

Moreover, the contrast-to-noise ratio (CNR) was also evaluated as follows:CNR*_k_*(*z*) = *C_k_*(*z*)/*σ*[log *N_k_*(*z*)],(5)
where *σ* indicates the standard deviation over the 10 repetitions.

Longitudinal resolution: the same phantom used in the previous measurement was employed, this time using both source-detector separations, 1.5 cm and 2.5 cm, respectively. The depth of the inclusion was fixed at 15 mm and its position along the source-detector axis (*x* coordinate) was varied from -40 mm to 40 mm in steps of 1 mm, *x* = 0 being the midpoint between source and short-distance detection. For each position, 10 repetitions were acquired with an integration time of 500 ms per wavelength, and the corresponding contrast *C_k_*(*x*) was calculated using Equation (4) with the *x* coordinate instead of the *z* coordinate.Depth selectivity: The capability of the device to separate absorption changes occurring in shallow vs. deep compartments was evaluated using bilayer liquid phantoms made of distilled water, Intralipid and black India ink. The thickness of the upper layer was fixed at 10 mm. First, a measurement was taken with the upper and lower compartments containing the same phantom (nominal optical parameters *µ_a_*_0_ = 0.15 cm^−1^, *µ_s_*_0_′ = 10 cm^−1^). Then, for a given absorption change Δ*µ_a_*, the absorption coefficient of the upper layer was increased by Δ*µ_a_* while keeping that of the lower layer fixed to *µ_a_*_0_. The contrast for the absorption change in the upper layer *C_k_*_,UP_(Δ*µ_a_*) was calculated as follows:

*C_k_*_,UP_(Δ*µ_a_*) = − log[*N_k_*(*µ_a_*_0_ +Δ*µ_a_*)/*N_k_*(*µ_a_*_0_)].(6)

An analogous procedure was then repeated for calculating the contrast in the lower layer *C_k_*_,DOWN_(Δ*µ_a_*), changing the absorption coefficient in the lower layer only. Finally, the depth selectivity *S* was evaluated as the ratio between the contrasts for the two layers:*S_k_*(Δ*µ_a_*) = *C_k_*_,UP_(Δ*µ_a_*)/*C_k_*_,DOWN_(Δ*µ_a_*).(7)

The values of Δ*µ_a_* ranged from 0.01 cm^−1^ to 0.1 cm^−1^ in steps of 0.01 cm^−1^. In all cases, the measurements consisted of 10 repetitions, with an integration time of 500 ms per wavelength.

#### 2.2.4. DCS Module Characterization

The characterization of the DCS module was performed according to the following procedures:Measurements at variable viscosity levels: the value of the Brownian diffusion coefficient *D_B_* for a phantom depends on the phantom properties according to the Einstein relation [29]:
*D_B_* = *k_B_T*/*6πηr*,(8)
where *k_B_* is the Boltzmann constant, *T* is the absolute phantom temperature, *η* its viscosity and *r* is the radius of the scattering particles. To test the ability of the device to follow *D_B_* variations, liquid phantoms with the same optical parameters (*µ_a_* ≅ 0.1 cm^−1^, *µ_s_*′ ≅ 10 cm^−1^) but different viscosities were produced [28]. To vary their viscosity, in addition to distilled water, Intralipid and black India ink, the phantoms also contained glycerol in concentrations ranging from 0% to 30% in steps of 10%. They were measured both individually and as part of bilayer phantoms. For the bilayer measurements, the glycerol concentration of the lower layer was kept at 0%, while that of the upper layer was increased from 10% to 30%. Each measurement consisted of 30 repetitions with 1 s integration time.

Stability: the same phantom and procedure used for evaluating the stability of the TD NIRS module were employed to assess the stability of the DCS component. The quantities measured were the Brownian diffusion coefficient *D_B_* and the coherence parameter *β* over the 14-h measurement.Reproducibility: the same phantom and procedure used in assessing the reproducibility of the TD NIRS module were employed to evaluate the reproducibility of the DCS component. The quantities measured were the Brownian diffusion coefficient *D_B_* and the coherence parameter *β*, calculating their CV over the days.DCS noise level: the noise level of the DCS module was assessed following the procedure described by Cortese et al. [30]. Briefly, a liquid phantom (*µ_a_* ≅ 0.05 cm^−1^, *µ_s_*′ ≅ 7 cm^−1^) was measured several times at different count rates. Each measurement consisted of 100 iterations of 1 s integration time, using all four detectors at a single source-detector separation *ρ* = 2.5 cm. Data were analyzed as described in [30] to obtain the dependence of the CV of *D_B_* on measurement duration, number of detection channels and count rate.

Finally, to complement the characterization of the hybrid device, a probe repositioning test was performed:To test the consistency of the instrument, a liquid phantom made of distilled water, Intralipid and black India ink (*µ_a_* ≅ 0.1 cm^−1^, *µ_s_*′ ≅ 10 cm^−1^) was measured five consecutive times, removing and repositioning the probe in between measurements. Each measurement consisted of 30 repetitions at 1 s integration time, and the inter-measurement CV of the following parameters was evaluated: *µ_a_*, *µ_s_*′, *D_B_*, and *β*.

### 2.3. In-Vivo Measurements

In-vivo measurements were carried out to test the ability of the hybrid device to monitor hemodynamic and microvascular perfusion changes during clinical protocols.

#### 2.3.1. Stepped Vascular Occlusion

To follow hemodynamic variations in response to gradual environmental changes, a stepped vascular occlusion protocol on four healthy subjects was implemented. After measuring their blood pressure, the volunteers sat with their arms lying at about the same height as the heart. A pneumatic cuff was placed on the left bicep, while the probe was placed on the left flexor carpi ulnaris muscle. A black auto-adhesive elastic bandage was used to fix the probe, ensuring its firm adherence to the skin to prevent motion artifacts during the measurement. The bandage is also effective for shielding ambient light, as confirmed through visual inspection of the DTOF background. The protocol consisted of 2 min baseline, 1 min light occlusion (30 mmHg below diastolic pressure), 1 min venous occlusion (at the midpoint between systolic and diastolic pressure), 1 min arterial occlusion (30 mmHg above systolic pressure) and 5 min of recovery after cuff deflation. The acquisition rate was set to 5 Hz.

Data were analyzed using the solution of the diffusion equation (for TD NIRS) and correlation diffusion equation (for DCS) for homogenous semi-infinite media. A 5-sample moving median filter was applied to the DCS autocorrelation curves before fitting, and the BFI was calculated assuming effective Brownian motion. From the absorption coefficient measured with TD NIRS, Beer’s law was used to extract HbO_2_ and HHb, which allowed us to then calculate tHb and StO_2_. Finally, a 5-sample moving average filter was applied to all signals for visualization.

#### 2.3.2. Pulsatility Measurements

To test the performance of the device at high acquisition rates, measurements of the pulsatile component of blood flow on one volunteer were also acquired. Two measurements were carried out on the brain and the forearm, respectively. For the former, the probe was placed on the forehead above the eyebrow, while, for the latter, it was placed on the flexor carpi ulnaris muscle. For this measurement, a different probe was used, with a single source-detector separation of 1.5 cm and no attenuation at the detection stage. The measurements lasted 2 min, with an integration time of 30 ms (33 Hz sample frequency).

Data were analyzed using the solution of the correlation diffusion equation for a homogeneous semi-infinite medium, and the BFI was calculated assuming effective Brownian motion. The power spectral density (PSD) of the BFI signal was computed using Welch’s method, with Hamming windows of 20 s and 50% overlap.

## 3. Results

The results of the phantom characterization and in-vivo measurements will be presented in this section.

### 3.1. Characterization Measurements Results

#### 3.1.1. BIP Protocol Results

Table 1 summarizes relevant parameters relative to the TD NIRS module for both laser sources. The responsivity and afterpulsing ratios show the same values for both detectors. Overall, the results are in line with other state-of-the-art TD NIRS devices [25].

The profiles of the IRFs for each source-detector pair are shown in Figure 2. As can be seen, all IRFs exhibit similar features. The falling edges are characterized by the double exponential decay typical of SiPMs. Minor secondary peaks are present around 2 ns after the main ones, likely originating from spurious reflections inside the fiber bundles.

The bottom plot of Figure 3a shows the DNL of the timing electronics for both detection channels after normalization to the mean value across time channels. Aside from a slight modulation over the time range, which is similar for both channels, the DNL of channel 1 (blue curve) also exhibits four sharp peaks and valleys distributed non-periodically over the time range. These features are likely due to electronic crosstalk between start and stop pulse signals and could, in principle, distort the reconstructed histograms. However, as shown in the top plot of Figure 3a, such distortions fall outside the range of a typical DTOF, so they do not affect the portion of the histogram that is relevant for data analysis. Additionally, their effect is very small: they do not appear in the DTOFs that are acquired during typical measurements (containing around 500k total photons) like those of Figure 3a, while they are barely visible in Figure 3b, which shows the sum of histograms reconstructed in a 13-h measurement.

The behavior of the IRF center of mass and FWHM over time was evaluated via a 12-h stability measurement. The position of the center of mass shifted forward over the first 6 h, after which it stabilized 20 ps (for channel 1) or 30 ps (for channel 2) after its initial value. On the other hand, the FWHM was much more stable, reaching stationarity in about 1 h for the laser source at 685 nm and immediately for the 830 nm laser source. Once stationarity had been reached, fluctuations of the FWHM were small, with the standard deviation being 1% of the mean value. As for absolute values, the FWHMs ranged from 190 ps to 220 ps, depending on the source-detector pair considered. In particular, for a given channel, the IRF at 830 nm was about 20 ps narrower than the one at 685 nm, while, for a fixed source, the IRF on channel 2 was about 10 ps narrower than the one on channel 1.

#### 3.1.2. MEDPHOT Protocol Results

Results for the accuracy and linearity at 830 nm for both source-detector separations are shown in Figure 4. Similar results are obtained for 685 nm (not shown). Panel (a) shows the values of the absorption coefficient measured for each phantom, plotted against the conventional ones (measured with the analogous device). Data points are well aligned, showing good absorption linearity, and the fact that they are disposed along the bisector line and generally fall within the ±3% error band indicates good accuracy as well. Similar considerations hold for the scattering coefficient, which is shown in panel (d). The small discrepancies between measured and conventional values are likely due to the difference in the IRF FWHM between the two instruments considered: indeed, the device presented here has wider IRFs compared to the one used for reference (~200 ps vs. ~160 ps). This leads to a less accurate reconstruction of the optical properties when the DTOFs are narrow, such as for short source-detector separations and high absorptions. Nevertheless, for the phantoms that are most representative of biological tissue (B series for scattering, series 2 to 5 for absorption), the mean absolute value of the relative error is 2.9%.

The coupling between absorption and scattering is shown in panels (b) and (c). From panel (b), it can be seen that, for each scattering series, save for the first two low-absorbing phantoms, the scattering coefficient measured is essentially independent of the absorption of the sample. Additionally, there is good agreement between the values measured at 1.5 cm (full markers) and 2.5 cm (hollow markers). Similarly, from panel (c), it can be seen that measured absorption is independent of scattering, and, again, there is good agreement between the two detection channels.

The results of the noise measurement are shown in Figure 5, which shows the decreasing trend of the CV of the absorption coefficient with the number of counts in the DTOF in a log–log scale. Linear fits (R^2^ = 0.98 for both wavelengths) yield slopes of −0.53 at 685 nm and −0.54 at 830 nm, in good agreement with the theoretical (i.e., Poisson-like) trend of the noise as the inverse square root of the detected intensity. To reach the conventional noise threshold of 3% CV at both wavelengths, 70,000 total counts are needed.

The behavior in time of the quantities measured by TD NIRS (*µ_a_*, *µ_s_*′ and photon count rate) and DCS (*D_B_* and *β*) was monitored via a 14-h stability measurement. The optical parameters retrieved at 1.5 cm source-detector separation took about 1 h to reach stationarity, while those at 2.5 cm source-detector separation only needed 15 min. The photon count rate, on the other hand, was much more stable, falling within 3% of the mean value at the regime for the whole duration of the measurement. Once stationarity had been reached, the optical parameters at both wavelengths and source-detector separations exhibited a CV of about 1%, while for the count rate, CV < 0.5%. Conversely, the DCS module did not exhibit any warm-up times, retrieving stationary values since the device startup. However, fluctuations were higher, with the CV for *D_B_* being 2.2% at the short source-detector distance and 4.6% at the long one, while those of *β* were 1.5% and 3.3%, respectively. The higher values of CV obtained at the long source-detector separation was due to the lower detected photon count rate. The absolute values of *µ_a_*, *µ_s_*′, *D_B_* and *β* measured at the two inter-fiber distances were the same.

Results of the reproducibility measurements showed good reproducibility for *µ_a_* and *µ_s_*′ at both inter-fiber distances, with the optical parameters measured each day being equal within one standard deviation and the inter-day CV being lower than 3%. Similar results were obtained for *D_B_* and *β*, which exhibited CV < 3.5%.

#### 3.1.3. nEUROPt Protocol Results

The results of the nEUROPt protocol measurements are shown in Figure 6. Panel (a) shows the contrast measured for an inclusion moving along the depth coordinate (*z*). For early time gates (blue curve), the contrast is relatively high when the inclusion is superficial but quickly decreases for deeper positions and vanishes around *z* = 12 mm. Late gates (red curve), on the other hand, exhibit lower contrast at shallow depths but are characterized by higher sensitivity to deeper inclusions. Indeed, the maximum contrast is reached about 4 mm below the surface, and it vanishes around *z* = 20 mm. The CW signal (yellow curve), obtained by summing the photons in all gates, represents a middle ground closer to the behavior of the early gates since they contain most of the total photons.

Panel (b) shows the contrast-to-noise ratio for the same measurement. Taking the arbitrary threshold CNR > 10, we can see that the contrast values shown in panel (a) were retrieved with sufficiently low noise to be considered reliable. Indeed, the CNR drops below 10 only when the contrast vanishes.

Panel (c) shows the contrast as a function of the position for an inclusion moving along the source-detector coordinate. The highest contrast is obtained for late gates, while early gates show the lowest. This agrees with the results shown in panel (a) since the inclusion is now located at z = 15 mm. Moreover, especially for late gates, the contrast is higher at *ρ* = 2.5 cm than at *ρ* = 1.5 cm, since for higher source-detector separations more photons reach the deep zones of the sample. The longitudinal resolution is given by the FWHM of the bell curves. Early gates provide slightly higher resolutions but at the cost of decreasing the absolute contrast.

Finally, the depth selectivity as a function of the absorption change is shown in panel (d). Being more sensitive to the bottom layer, late-arriving photons allow higher selectivity than early ones. Similarly, by increasing the source-detector separation, the collected photons have, on average, travelled deeper inside the sample so that the selectivity is higher at *ρ* = 2.5 cm than at *ρ* = 1.5 cm.

#### 3.1.4. DCS Module Characterization Results

The Brownian diffusion coefficient measured for the homogenous phantoms is plotted in Figure 7 as a function of the phantom glycerol concentration. The decreasing trend of *D_B_* with the glycerol concentration is correctly retrieved. The values measured at the two source-detector separations are similar, being slightly higher at the short distance for lower viscosities.

Figure 8a shows the Brownian diffusion coefficient measured for the bilayer phantoms as a function of the upper layer thickness. Since data were analyzed assuming a homogenous medium, the resulting *D_B_* lies between that of the two layers. In particular, as the upper layer thickness is increased, the measured *D_B_* gets closer and closer to that of the superficial layer, as expected. This effect is stronger at the short source source-detector separation since photons collected closer to the source do not reach as deep into the sample. As such, especially for lower thicknesses, the *D_B_* measured at *ρ* = 2.5 cm is closer to that of the bottom layer. Plotting *D_B_* as a function of the glycerol concentration of the upper layer instead (the glycerol concentration in the lower layer is kept constant, panel (b)), the same decreasing trend of Figure 7 is again observed, but with enhanced differences between the two detection channels due to the inhomogeneity of the phantom.

The results of the noise measurement for the DCS module are reported in Figure 9. The bi-logarithmic plot in panel (a) shows the dependence of the CV of *D_B_* on the count-rate, using all four detectors. The lines corresponding to the different measurement durations all have the same slope, which is close to the theoretical one (dashed red line) [31]. To reach the arbitrary threshold of 5% CV, a 40 kHz count-rate is needed for 1 s measurements. The results obtained for the dependence of the CV on the number of detectors and on measurement duration similarly show good agreement with the theoretical trend (not shown). Panel (b) shows the trend of the square root of the sum of the variance of the autocorrelation curve at each time bin. Being directly related to the raw DCS signal, this quantity is more representative of the noise level of the apparatus [30].

The results of the probe repositioning measurement showed no meaningful differences in the values of the optical parameters measured after each reposition. Indeed, measured values are equal within one standard deviation, and the inter-measure CV is lower than 1%. Similar results hold for *D_B_* and *β*, whose CV is <1.5%.

### 3.2. In-Vivo Measurements Results

#### 3.2.1. Stepped Vascular Occlusion Results

The time traces of the hemodynamic parameters of one representative subject during the stepped occlusion protocol are shown in Figure 10. After the initial baseline, when the cuff is inflated below diastolic pressure, all signals stay constant since no occlusion is being induced. When the pressure is increased to induce venous occlusion, both HbO_2_ and HHb increase, while StO_2_ stays constant; the BFI decreases slightly. Then, when the cuff is inflated further to occlude the artery, HbO_2_ decreases while HHb keeps increasing so that tHb stays constant and StO_2_ decreases. The BFI drops abruptly at the beginning of the arterial occlusion. When the cuff is released, HbO_2_, StO_2_ and BFI exhibit hyperemic peaks. Similarly, HHb and tHb show an undershoot before returning to baseline. These results are in accordance with the literature [14].

#### 3.2.2. Pulsatility Measurements Results

The top plots of Figure 11 show the BFI signal over the first 10 s of measurement for both the brain (panel (a)) and the forearm (panel (b)). Since the brain is less superficial than forearm muscles, the pulsatile wave is more visible in the latter, with an amplitude about 10 times greater. Nevertheless, the peak at the cardiac frequency is visible in both spectra (bottom plots), together with its higher-order harmonics. These results show the ability of the DCS module to detect the fast changes associated with the BFI signal.

## 4. Discussion

In this work, we have presented a new hybrid TD NIRS and DCS device and characterized it according to established protocols.

Low-level characterization of the TD NIRS module via the BIP protocol showed results in line with other state-of-the-art TD NIRS instruments [25].

The behavior of the system over time was assessed both in terms of stability during a long measurement and reproducibility over several days. The results showed short warm-up times and excellent stability of the measured quantities, guaranteeing the reliability of the device during long measurement sessions. Moreover, the high reproducibility of the device makes it suitable for clinical studies.

Measurements on calibrated phantoms showed good linearity and accuracy for the TD NIRS module, proving its ability to detect variations of hemodynamic parameters without distortions, as well as to accurately retrieve their absolute values. As for the DCS module, measurements on liquid homogeneous phantoms demonstrated its ability to correctly detect variations in the dynamic flow of scatterers. Assessing the accuracy of DCS measurements is still an open question, which is outside the scope of this work.

The ability of the device to probe different tissue layers has been evaluated as well. For TD NIRS, on top of using the double-distance approach, time gating offers some depth discrimination even when using a single detection channel. Nevertheless, better results could be obtained by adopting more refined data analysis models, which, however, are not part of the characterization protocols adopted. For DCS, the presence of two measurement channels working at different inter-fiber distances allows some depth discrimination even when using a simple homogenous model for data analysis.

Finally, in-vivo measurements showed results in accordance with the literature, validating the device for clinical use. When using both detection channels, as in the case of the stepped occlusion protocol, it is necessary to attenuate the signal at the short source-detector separation to avoid the pile-up effect in the TD NIRS signal. Thus, only a few photons are detected for DCS as well, imposing a limit on the maximum achievable sampling frequency. We stress that this is a “soft” limit due entirely to the low DCS signal-to-noise ratio rather than a “hard” one originating from hardware limitations: indeed, the software correlator does not impose any limitations to the sampling frequency that can be achieved in post-processing.

## 5. Conclusions

We presented a new hybrid multi-channel TD NIRS and DCS device. The system improves upon another instrument we previously developed [15], featuring two separate detection channels for both TD NIRS and DCS, which allows for better depth discrimination in layered tissues [20,21] and is able to operate at sampling frequencies of up to 50 Hz. The device was characterized according to established protocols and in-vivo on volunteers. The results demonstrated the capability of the instrument to make robust and reliable measurements, making it suitable for clinical applications such as brain and skeletal muscle hemodynamic monitoring. Indeed, on top of traditional functional state evaluation, the high temporal resolution of the device enables reliable detection of pulsatile signals [13] and fast muscle activations during exercise protocols such as cycling [6].

In the future, data analysis models that take full advantage of the double source-detector separation will be implemented for more accurate and reliable results on multi-layered samples. Additionally, moving the attenuation stage from the probe to the internal detection optics of the TD NIRS module, albeit offering less flexibility, would allow faster acquisition rates at a short distance. Indeed, when the intensity is not attenuated, the high photon count rate achieved allows us to detect the fast pulsatile signal at 1.5 cm. Alternatively, the inclusion of a beam splitter to the TD NIRS injection optics would enable two separate injection spots for both TD NIRS and DCS, allowing simultaneous multi-site monitoring at a single source-detector separation. While reducing depth selectivity, such an alternative configuration allows investigation of regional differences in the tissue under examination.

## Figures and Tables

**Figure 1 sensors-24-07375-f001:**
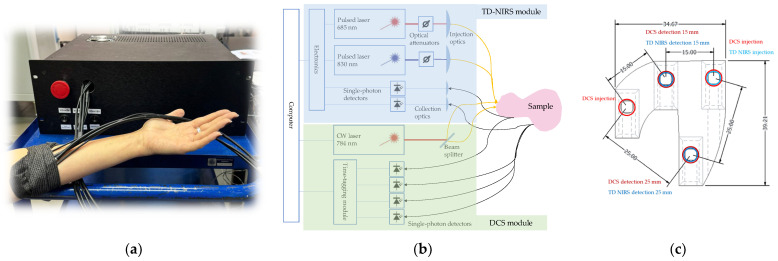
(**a**) Picture of the setup; (**b**) block diagram of the device; (**c**) sketch of the optical probe seen from the bottom (units are in mm).

**Figure 2 sensors-24-07375-f002:**
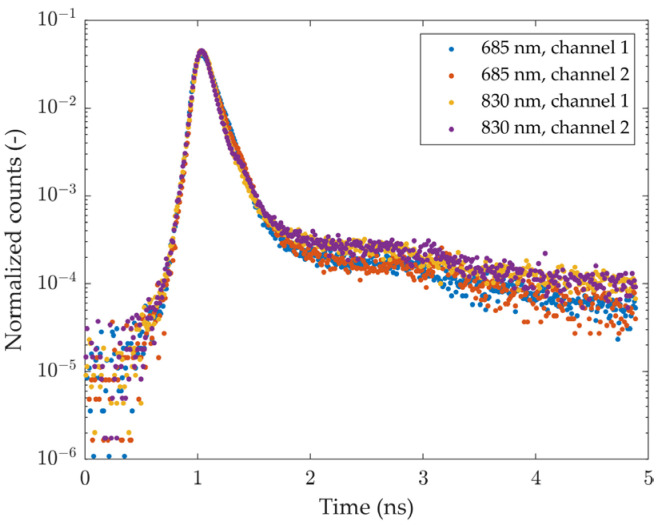
Overlapped IRFs for each source-detector pair. The curves had their backgrounds subtracted, were normalized to their respective areas and were shifted so that all their peaks coincide.

**Figure 3 sensors-24-07375-f003:**
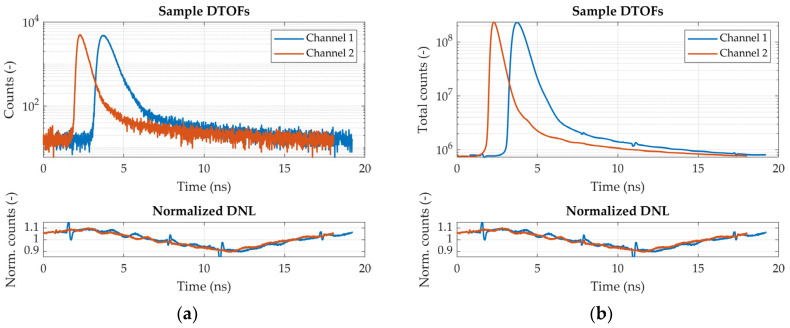
Sample DTOF curves and DNL (**a**) DTOF curves resulting from 500 ms integration time (**top plot**) compared to the DNL (**bottom plot**); (**b**) DTOF curves obtained by summing all the curves from a 13-h measurement (**top plot**) compared to the DNL (**bottom plot**).

**Figure 4 sensors-24-07375-f004:**
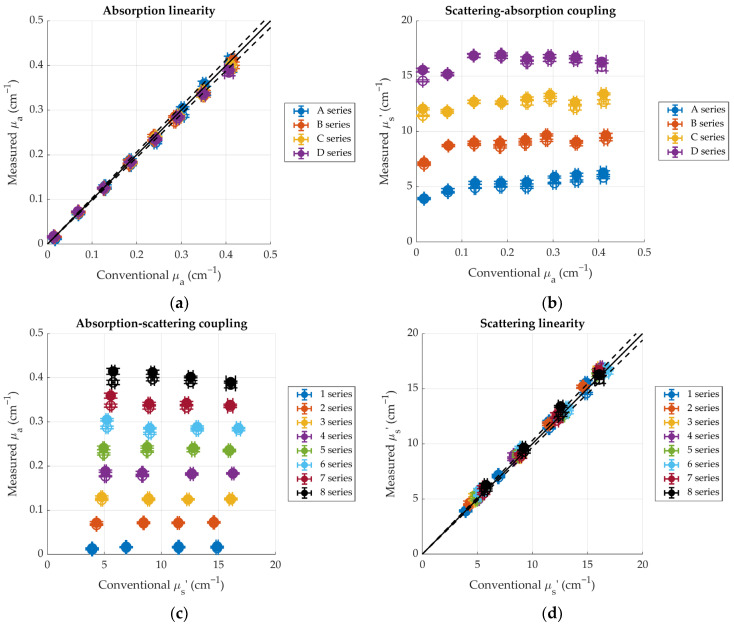
Linearity plots for the optical parameters of the MEDPHOT phantoms at 830 nm. Full markers represent the values measured for an inter-fiber distance *ρ* = 1.5 cm, hollow markers are for *ρ* = 2.5 cm. Error bars represent standard deviations over 10 repetitions. (**a**) Measured absorption vs. conventional absorption; (**b**) measured scattering vs. conventional absorption; (**c**) measured absorption vs. conventional scattering; (**d**) measured scattering vs. conventional scattering. Different colors represent different scattering (panels (**a**,**b**)) or absorption (panels (**c**,**d**)) series. Panels (**a**,**d**) show 1:1 lines (solid) and ±3% lines (dashed).

**Figure 5 sensors-24-07375-f005:**
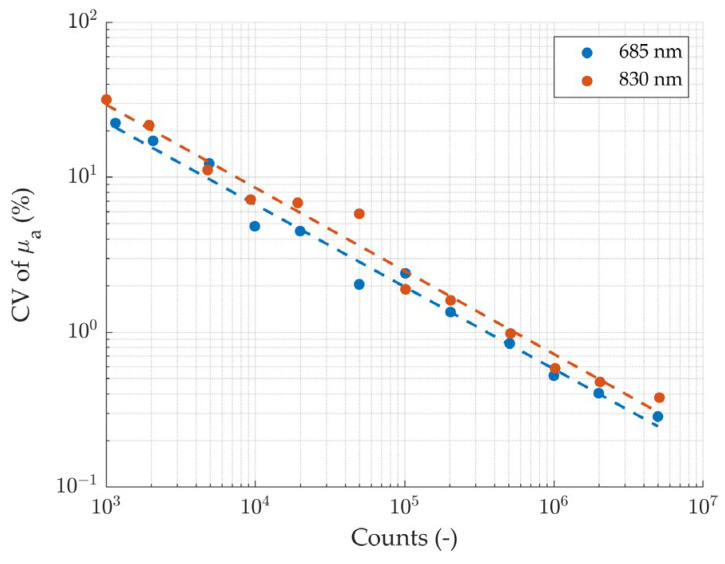
CV of the absorption coefficient over 10 repeated measurements of the same phantom at a source-detector separation *ρ* = 2.5 cm as a function of the photon count of the DTOF (log-log scale). Dashed lines represent linear fits.

**Figure 6 sensors-24-07375-f006:**
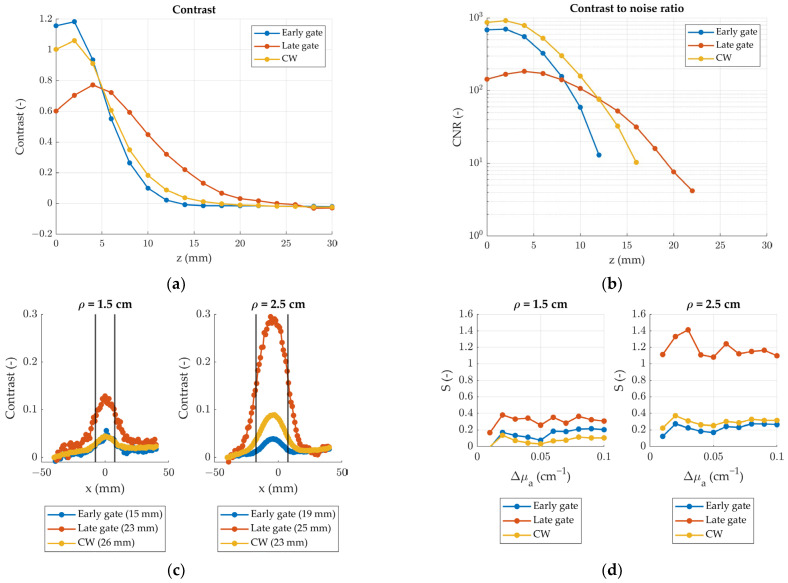
Relevant nEUROPt parameters of the device considering early time gates (500 ps to 1000 ps), late time gates (2000 ps to 2500 ps) and CW signal (summing all gates). (**a**) Contrast as a function of the inclusion depth for a source-detector separation *ρ* = 2.5 cm; (**b**) contrast to noise ratio as a function of the inclusion depth for *ρ* = 2.5 cm; (**c**) contrast as a function of the longitudinal position of the inclusion. Black vertical lines represent the source and detector positions, numbers in legend are the FWHM; (**d**) depth selectivity as a function of the absorption coefficient variation.

**Figure 7 sensors-24-07375-f007:**
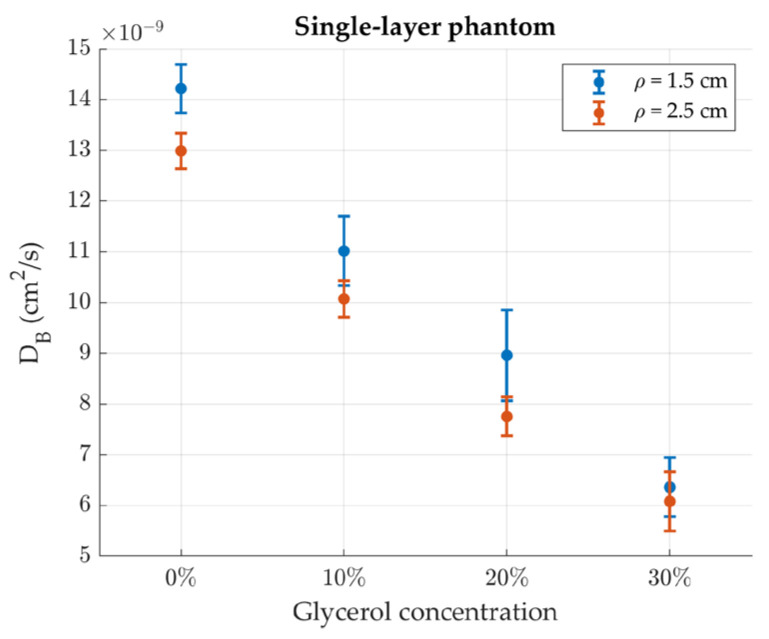
Effective Brownian diffusion coefficient of the homogeneous phantoms as a function of the glycerol concentration. Error bars represent standard deviations over 30 repetitions.

**Figure 8 sensors-24-07375-f008:**
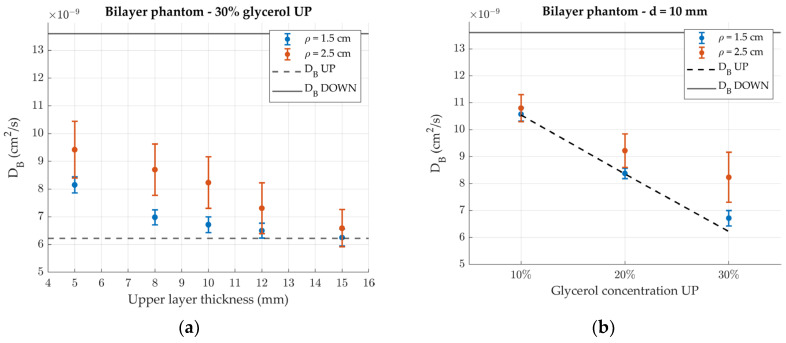
Effective Brownian diffusion coefficient of the bilayer phantoms. Error bars represent standard deviations over 30 repetitions. Black lines represent the values of *D_B_* in the two layers. (**a**) Measured *D_B_* of the phantom with 30% glycerol concentration in the upper layer as a function of its thickness. (**b**) Measured *D_B_* of the different phantoms for an upper layer thickness d = 10 mm.

**Figure 9 sensors-24-07375-f009:**
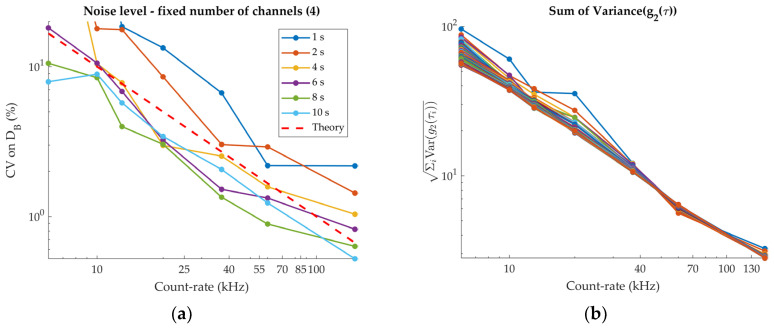
Noise level of the DCS module. (**a**) CV of *D_B_* for a fixed number of detection channels (4) as a function of the count-rate, for different measurement durations. The dashed red line represents the theoretical trend, and only its slope is significant. (**b**) Sum of the variance of the autocorrelation curves as a function of the count-rate. Each line represents a different measurement duration (from 1 s up to 100 s).

**Figure 10 sensors-24-07375-f010:**
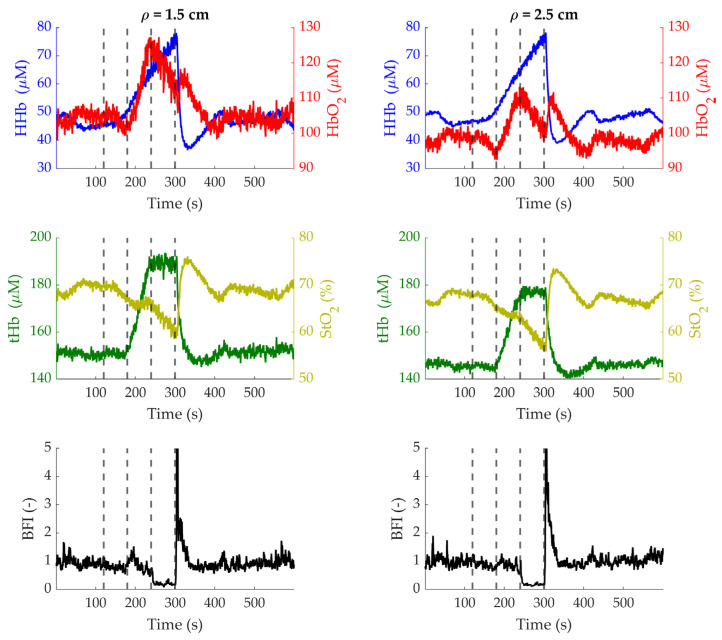
Time courses of the hemodynamic parameters measured at *ρ* = 1.5 cm (**left column**) and *ρ* = 2.5 cm (**right column**) during the stepped occlusion on subject 2. Top row: HbO_2_ (red), HHb (blue); middle row: tHb (dark green), StO_2_ (light green); bottom row: BFI. Dashed vertical lines represent the start of each step and the cuff release at the end of the occlusion.

**Figure 11 sensors-24-07375-f011:**
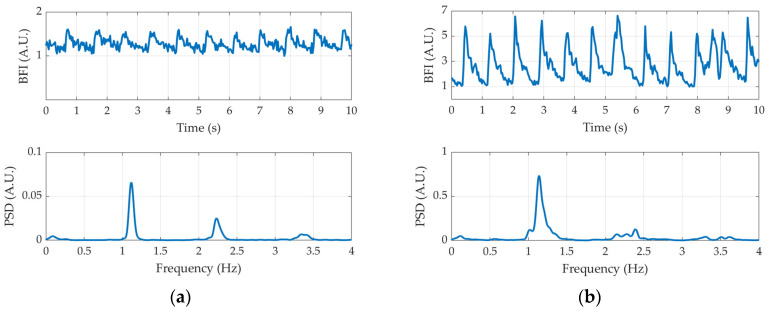
Time traces of the BFI signals (top plots) and their PSDs (bottom plots). (**a**) Brain; (**b**) forearm.

**Table 1 sensors-24-07375-t001:** BIP synthetic descriptors of the device.

Synthetic Descriptor	Value	
	685 nm	830 nm
Illuminated area on sample surface	2.89 mm^2^	2.89 mm^2^
Maximum power delivered to sample	3.32 mW	2.58 mW
Responsivity	1.44 × 10^−8^ m^2^ sr	0.90 × 10^−8^ m^2^ sr
Afterpulsing ratio	2.4%	5.0%

## Data Availability

The original data presented in this study are openly available in Zenodo at https://doi.org/10.5281/zenodo.13922081 (accessed on 10 November 2024).

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
