# Peer review of "Fast Multi-Distance Time-Domain NIRS and DCS System for Clinical Applications"

_sensors, 2024, doi:10.3390/s24227375_

Round 1

Reviewer 1 Report

Comments and Suggestions for Authors

1. The conclusion section is rather brief, merely summarizing the experimental results without offering insight into future research directions or potential improvements to the device. It is recommended to expand the conclusion section by discussing future work directions, the clinical application prospects of the device, and possible technological improvements, in order to enhance the overall forward-looking nature of the study.

2. In the in vivo experiments, although several tests were designed (such as pulsatility measurements and vascular occlusion experiments), there was no detailed explanation of how potential confounding factors (e.g., ambient light interference, motion artifacts) were controlled. Increasing control measures for the experimental environment and participant conditions could improve the reliability of the data.

3. It appears that only a small number (4) of healthy volunteers were used for testing in the in vivo experiments. Given the individual differences in human hemodynamic parameters, such a small sample size is insufficient to ensure the representativeness of the data. To derive statistically significant conclusions, it is recommended to increase the sample size in the in vivo experiments, particularly for studies measuring microcirculation and cerebral hemodynamics, where individual variability can be substantial. A sample size of at least 20–30 subjects would better assess the effectiveness and reliability of the device in real-world applications.

4. The results section mentions that the data errors in the linearity tests for optical absorption and scattering coefficients are "within ±3%," but some data points in Figure 4 actually exceed this range, particularly in cases of low absorption and high scattering coefficients. It is suggested to recalculate and explain the data errors in the charts, clarifying why the errors exceeded the expected range in some tests and how future testing designs could be improved to reduce these discrepancies.

In summary, I think this work can be publishable after major revision.

Reviewer 2 Report

Comments and Suggestions for Authors

The presented work is performed at a good scientific and technical level, it describes in detail the parameters of the installation, the methods and protocols of the studies, as well as the results of the experiments. The authors generalized the results obtained, made conclusions about its practical applicability and ways of further improvement.

It can be concluded that this work is of interest to scientists and specialists conducting both fundamental and applied research in the field of biomedicine.

There are no significant comments on the work, the style and clarity of the material leave no questions. Perhaps the addition of photographs of the installation, probe and samples to the text would additionally enliven and decorate the publication. However, this comment is completely not fundamental.

As the result, I believe that this work is worth publishing in the journal Sensors and can be accepted in its current form.

Reviewer 3 Report

Comments and Suggestions for Authors

This manuscript presents a hybrid system for Time-Domain Near-Infrared Spectroscopy (TD-NIRS) and Diffuse Correlation Spectroscopy (DCS), designed for fast, multi-distance clinical applications with an operation rate up to 50 Hz. The device had gone through comprehensive testing on calibrated phantoms to ensure accuracy, stability, and reproducibility. In-vivo tests included vascular occlusion and pulsatility measurements, showing consistent performance in real settings. This work fits the scope of this journal well and should attract potential readers. However, revisions are necessary for publication in Sensors, as evidenced by the following points.

1.      The introduction section elaborates on the applications and advantages of diffusion optics in biomedical fields, specifically highlighting the advantages of combining TD-NIRS and DCS for a more comprehensive tissue characterization and citing the benefits of having simultaneous measurements of blood flow and tissue oxygenation. However, the authors should explain their hybrid system in more detail and compare advantages and disadvantages between the system in their previous work, especially in the last paragraph in the introduction, as this is the main novelty of the paper.

2.      Following the previous point, there has been development of hybrid TD-NIRS and DCS systems similar to the one described in the paper (such as Journal of Applied Physiology, 127(5), 1328–1337.). Including references from these works would demonstrate the advancements and challenges in hybrid TD-NIRS and DCS systems, highlighting the authors' contributions in a broader research framework.

3.      The authors show a block diagram and the sketch of the probe in Fig. 1. It would be more beneficial if the authors could also include a panel of the photo of their actual setup which is helpful for the readers to understand.

4.      The authors explain that the improved depth selectivity in the late gate at a source-detector separation of 2.5 cm results from the behavior of late-arriving photons, which travel deeper into the tissue before reaching the detector. To strengthen this argument, they could perform a quantitative analysis comparing depth selectivity at various depths using both time gates and source-detector separations. This would provide numerical evidence of how depth selectivity varies with both gating and separation.
